# Microbiome Patterns in Matched Bile, Duodenal, Pancreatic Tumor Tissue, Drainage, and Stool Samples: Association with Preoperative Stenting and Postoperative Pancreatic Fistula Development

**DOI:** 10.3390/jcm9092785

**Published:** 2020-08-28

**Authors:** Melanie Langheinrich, Stefan Wirtz, Barbara Kneis, Matthias M. Gittler, Olaf Tyc, Robert Schierwagen, Maximilian Brunner, Christian Krautz, Georg F. Weber, Christian Pilarsky, Jonel Trebicka, Abbas Agaimy, Robert Grützmann, Stephan Kersting

**Affiliations:** 1Department of Surgery, University Hospital of Erlangen, 91054 Erlangen, Germany; barbara.kneis@web.de (B.K.); matthiasgittler@yahoo.de (M.M.G.); maximilian.brunner@uk-erlangen.de (M.B.); christian.krautz@uk-erlangen.de (C.K.); georg.weber@uk-erlangen.de (G.F.W.); Christian.pilarsky@uk-erlangen.de (C.P.); robert.gruetzmann@uk-erlangen.de (R.G.); stephan.kersting@uk-erlangen.de (S.K.); 2Department of Internal Medicine 1, University Hospital of Erlangen, 91054 Erlangen, Germany; stefan.wirtz@uk-erlangen.de; 3Translational Hepatology, Department of Internal Medicine I, University Clinic Frankfurt, 60590 Frankfurt, Germany; Olaf.Tyc@kgu.de (O.T.); Robert.Schierwagen@kgu.de (R.S.); Jonel.Trebicka@kgu.de (J.T.); 4Department of Pathology, University Hospital of Erlangen, 91054 Erlangen, Germany; abbas.agaimy@uk-erlangen.de

**Keywords:** microbiome, pancreatic surgery, postoperative complications, pancreatic cancer, 16S rRNA sequencing

## Abstract

Postoperative complications after pancreatic surgery are still a significant problem in clinical practice. The aim of this study was to characterize and compare the microbiomes of different body compartments (bile duct, duodenal mucosa, pancreatic tumor lesion, postoperative drainage fluid, and stool samples; preoperative and postoperative) in patients undergoing pancreatic surgery for suspected pancreatic cancer, and their association with relevant clinical factors (stent placement, pancreatic fistula, and gland texture). For this, solid (duodenal mucosa, pancreatic tumor tissue, stool) and liquid (bile, drainage fluid) biopsy samples of 10 patients were analyzed using 16s rRNA gene next-generation sequencing. Our analysis revealed: (i) a distinct microbiome in the different compartments, (ii) markedly higher abundance of *Enterococcus* in patients undergoing preoperative stent placement in the common bile duct, (iii) significant differences in the beta diversity between patients who developed a postoperative pancreatic fistula (POPF B/C), (iv) patients with POPF B/C were more likely to have bacteria belonging to the genus *Enterococcus*, and (v) differences in microbiome composition with regard to the pancreatic gland texture. The structure of the microbiome is distinctive in different compartments, and can be associated with the development of a postoperative pancreatic fistula.

## 1. Introduction

Pancreatic cancer (PC) is currently the seventh leading cause of cancer-related mortality worldwide, and is projected to become the second leading cause of cancer-related death by 2030 [1]. Radical surgery still affords the only chance of a potential cure for PC. Despite the improvement in surgical techniques, new devices and technologies, and intensive care management, postoperative complications and morbidity remain a challenge after pancreatic surgery. Postoperative major complications affect the prognosis and outcome of patients. Data from the prospective StuDoQ Pancreas registry of the German Society of General and Visceral Surgery yield a rate of 15% (12–19%) for major complications (Clavien-Dindo Classification (CDC) grades 3b and 4) in elective pancreatoduodenectomy [2]. A postoperative pancreatic fistula (POPF) is the most dangerous complication, and leads to a risk of sepsis, prolonged hospitalization, increased morbidity and mortality, and higher hospital costs [3,4,5,6]. Recently, the microbiome of the gut has not only been linked to anastomotic leakages after colorectal surgeries [7], but also to surgical complications in pancreatic operations [8].

During the last decade, extensive research has been done on the human microbiome and its role in health and disease [6,9]. Technical advances in detection (e.g., next-generation sequencing (NGS)) have changed our understanding of the structure and function of the microbiome, and the influence of the microbiome on immunity and cancer [10]. Altered local microbiota compositions in different cancer tissues may impact cancer therapy response [5]. In line with these concepts, compelling research has demonstrated that the gut microbiome is associated with impaired anastomotic healing in colorectal cancer (CRC), or higher postoperative complication rates in pancreatic surgery [8,11]. *Enterococcus faecalis*, for instance, impairs anastomotic healing in CRC via enhancing collagen-degrading activity and activating intestinal tissue matrix metalloproteinase 9 (MMP9) [7,12]. Furthermore, several studies have indicated that compartments formerly viewed as sterile, such as the bile or the pancreas, harbor their own autochthonous site-specific microbiome [13,14,15]. Translocation of bacteria into the pancreas may occur either from the duodenum via the biliary/pancreatic duct, or through the circulatory system [16,17]. Therefore, preoperative biliary stenting, as well as drains, could influence the composition of the pancreatic microbiome. Emerging preclinical data support that the microbiome can influence tumor progression and therapeutic responses through several pathways, such as through inflammation, immunity, and metabolism [13,18]. The molecular basis of this regulation is still being elucidated, and disputes exist as to whether the microbiota act directly in the cancer-initiating or progression cascade, or as a mediator of other stimuli such as inflammation. However, to date, there have been only a few studies of the microbiome within the bile duct, or in pancreas lesions in suspected cancer. The aim of this explorative study was to characterize and to compare the microbiome of different body sites (bile duct, duodenal mucosa, pancreatic tumor lesion, postoperative drainage fluid, and fecal samples) in patients undergoing pancreatic surgery for suspected pancreatic cancer, and to correlate these with clinical parameters.

## 2. Experiment

### 2.1. Study Design

This prospective, observational trial was approved by the Ethics Committee of the University of Erlangen, Germany (No 451_18B). Patients undergoing pancreatic surgery were screened for eligibility for study participation. Patients with neoadjuvant therapy, antibiotic therapy 4 weeks prior to surgery, patients with ulcerative colitis, Crohn’s disease, chronic pancreatitis, or another disease significantly affecting gastro-intestinal function were excluded. We performed an exploratory study on the first patients with pancreatic head resection in order to characterize the microbiome in different compartments related to the pancreas, and to see if the local microbiome is associated with the development of postoperative complications under different clinical conditions. Each patient received a standardized single shot of a 3rd generation cephalosporine and metronidazole about 30 min before the surgical procedure. In total, 50 samples from 10 patients were obtained. Seven patients underwent preoperative stent placement for biliary obstruction, six patients via ERCP and one patient received an external PTCD two days prior to surgery. An overview of the study workflow is depicted in Figure 1.

### 2.2. Sample Processing and DNA Purification

Stool samples were collected and stabilized the day before surgery (stool pre-op) and postoperative (stool post-op) on day 5–7 using the Omnigene Gut system (DNA Genotek, Ottawa, ON, Canada), and were stored at −80 °C until DNA extraction. DNA was extracted from the stool using the PSP Stool DNA stool kit, according to the specifications of the manufacturer (Invitek Molecular, Berlin, Germany). Tissue specimens of tumor tissue and duodenal tissue were collected immediately after resection, suspended in RNA later buffer, and stored at −80 °C. Bile fluid (500 µL) was collected intraoperatively after transection of the common bile duct through a sterile aspiration catheter, and immediately stored at −80 °C. Drainage fluid was collected postoperatively on days 3–5. DNA from tumor tissue, duodenal tissue, bile, and the drainage fluid was extracted using the Qiamp Microbiome Kit (Qiagen, Hilden, Germany), according to the manufacturer’s recommendations. DNA was subsequently quantified using a Qbit device (Thermo Fisher Scientific, Waltham, MA, USA). The V3+4 region of the 16S rRNA gene was amplified using 10 ng of bacterial template DNA with degenerate region-specific primers (341F: 5′-ACTCCTACGGGAGGCAGCAG-3′; 806R: 5′-123 GGACTACHVGGGTWTCTAAT-3′) containing barcodes and Illumina flow cell adaptor sequences [19] in a reaction consisting of 25 (stool) or 35 (tissue) PCR cycles (98 °C 15 s, 58 °C 20 s, 72 °C 40 s) using the NEBNext Ultra II Q5 Master Mix (New England Biolabs, Ipswich, MA, USA). Amplicons were purified with Agencourt AMPure XP Beads (Beckmann Coulter, Brea, CA, USA), normalized, and pooled before sequencing on an Illumina MiSeq device using a 600-cycle paired-end kit and the standard Illumina HP10 and HP11 sequencing primers. For bioinformatic processing, the terminal 15 bases of both forward and reverse reads were removed, before merging and quality filtering using the fastq mergepairs and fastq filter options from Usearch 10 [20]. Subsequently, merged fastq files were demultiplexed and trimmed using Cutadapt [21], 16S the Uparse [22], and Sintax [23] algorithms within Usearch using the silva 16S rRNA database (v123).

### 2.3. Statistical Analyses

The MicrobiomeAnalyst platform [24,25] was used to calculate alpha and beta diversities, and to compare the relative abundance of taxa. Linear discriminant analysis effect size (LEfSe) [26] was used to discover the key microbial taxa associated with the different compartments. A *p*-value of <0.05 was considered statistically significant.

## 3. Results

### 3.1. Patient Characteristics

We performed the analysis on the first patients with pancreatic head resection for suspected cancer. The clinical characteristics of the patients are summarized in Table 1.

### 3.2. Bacterial Composition at the Phylum and Genus Levels

We obtained patient-matched normal (bile fluid, duodenal tissue, stool preoperative/postoperative and postoperative drainage fluid) and pancreatic tumor tissue samples from 10 patients. The profile of bacterial DNA and the microbiome in the samples differed from those found in the quality controls (mock community, water). At the phylum level, the most abundant phyla in all samples were *Firmicutes* and Proteobacteria, followed by *Bacteroidetes, Actinobacteria*, *Verrucomicrobia* and *Fusobacteria*, to different degrees (Figure 2).

The phyla Bacteroidetes was significantly increased in the gut before and after operation, compared with those in the other compartments when evaluated with the univariate method (Mann-Whitney/Kruskal-Walis Test: *p* < 0.05 and FDR < 0.05).

At the genus level, the five most predominant genera in bile fluid were *Enterococcus, Streptococcus, Escherichia Shigella, Veilonella* and *Enterobacter*. In the duodenal group, the distribution was *Enterococcus, Enterobacter, Fusobacterium, Akkermansia* and *Veilonella*. In the pancreatic tumor samples *Enterococcus, Enterobacter, Fusobacterium, Barnesiella* and *Akkermansia* dominated. The predominant genera in the gut were *Bacteroides*, *Escherichia_Shigella*, *Clostridium_XlVa*, *Faecalibacterium* and *Enterobacter*. The drainage fluid mainly harbored *Enterococcus, Staphylococcus, Escherichia_Shigella, Streptococcus* and *Enterobacter* (Figure 2).

### 3.3. Identification of Key Taxa Associated with Different Samples

Linear discriminant analysis (LDA) coupled with effect size measurements (LEfSe) was applied to determine key taxa that were differentially represented in the different analyzed compartments. A total of 31 key genera were identified at the genus level (Figure 3). The eight key taxa in the tumor tissues were *Barnesiella* (LDA score 5.58, *p* = 0.027), *Blautia* (LDA score 5.39, *p* = 0.002), *Microbacterium* (LDA score 5.31, *p* = 0.00007), *Norcardia* (LDA score 5.04, *p* = 0.003), *Stenotrophomonas* (LDA score 5.03, *p* = 0.001), *Ruminococcus2* (LDA score 4.84, *p* = 0.001), *Ochrobactrum* (LDA score 4.67, *p* = 0.004), and *Collinsella* (LDA score 4.44, *p* = 0.01) (Figure 3).

### 3.4. Bacterial Richness and Diversity in the Samples

To estimate the overall richness and diversity of the bacterial communities, the alpha diversity indices were analyzed. We compared the Observed (richness), Chao1 (richness) and Shannon (evenness and richness) indices between the different compartments at the genus level. The overall structure of the microbiota in the microhabitats was significantly different, which was estimated by the Observed index (*p*-value: 4.9602 × 10^−7^; [ANOVA] F-value: 10.29), Chao1 index (*p*-value: 3.6826 × 10^−6^; [ANOVA] F-value: 8.716), and Shannon index (*p*-value: 0.0066998; [ANOVA] F-value: 3.6101).

Since it was unknown whether the microbial ecology of any of the upper alimentary sites investigated would resemble the tumor microbiome, we examined differences of the microbiome structure between the different compartments: gut (stool preoperative), duodenal mucosa, bile fluid and tumor tissue. The alpha diversity indices indicated that tumor tissue has a higher alpha-diversity compared with bile. However, there was not a clear differentiation between duodenal mucosa and pancreatic tumor tissue at the genus level. The gut (stool) microbiome had the highest alpha-diversity (Figure 4).

Moreover, beta-diversity analysis was performed through PCoA and NMDS analysis, based on the Bray–Curtis index and Jensen–Shannon divergence at the genus level. The analysis revealed that the overall structure of the microbiota in the different compartments was significantly different: PCoA Jensen–Shannon [PERMANOVA] F-value: 4.5694; R-squared: 0.28976; *p*-value < 0.001 (Figure 4 and Appendix A).

### 3.5. Relative Abundance and Associated Microbiome Profiles with Clinical Conditions

To better understand the relationships between clinical conditions and the impact of the microbiome, we analyzed three variables: preoperative stent placement, POPF B/C formation, and gland texture.

#### 3.5.1. Stent Placement in the Bile Duct

We first evaluated if stent placement in the common bile duct influences the microbiome of the different compartments (Figure 5). At the genus level, the five most predominant genera in bile fluid without stent placement were *Veilonella* (49%), *Escherichia_Shigella* (15%), *Enterococcus* (7%), *Enterobacter* (5%) and *Clostridium_sensu* (5%). In patients with stent placement, *Enterococcus* (32%), *Streptococcus* (23%), *Escherichia_Shigella* (20%), *Veilonella* (10%) and *Enterobacter* (5%) were found. In the duodenal group without stenting, the distribution was *Fusobacterium* (23%), *Enterococcus* (15%), *Clostridium_sensu* (14%), *Veilonella* (10%) and *Akkermansia* (9%), and in patients with stent placement it was *Enterococcus* (27%), *Enterobacter* (25%), *Bacteroides* (8%), *Akkermansia* (8%) and *Lactobacillis* (6%). In the no stenting tumor group we found *Fusobacterium* (22%), *Barnesiella* (17%), *Akkermansia* (13%), *Escherichia_Shigella* (8%), *Enterobacter* (5%), *Microbacterium* (5%) and *Clostridium_XIVa* (5%), and in patients with stenting we found *Enterococcus* (27%), *Enterobacter* (23%), *Akkermansia* (7%), *Bacteroides* (5%), *Norcardia* (4%) and *Fusobacterium* (4%). Due to the small group size (no stent placement patients *n* = 3), there were no statistically significant differences found, but all patients with stent placement had a higher abundance of *Enterococcus*.

#### 3.5.2. Postoperative Complications (POPF B/C)

Next, we divided the patients in two groups: one with postoperative pancreatic fistulas (POPF B/C *n* = 3), and the no POPF group (*n* = 7) (Figure 6). When comparing these two groups we observed at phylum level a higher abundance of Firmicutes in the bile (no POPF 48% vs. POPF 86%, mainly *Enterococcus*), duodenal tissue (no POPF 28% vs. POPF 72%; genus level no POPF *Enterococcus* 48% and *Veilonella* 40% vs. POPF *Enterococcus* 84% and *Lactobacillus* 11%) and pancreatic tumor tissue (no POPF 11% vs. POPF 77%), while in the gut the percentage of *Firmicutes* was not different (no POPF 35% vs. POPF 38%). Drainage fluid harbored in the no POPF group the phyla *Firmicutes* (52%), *Proteobacteria* (23%) and *Actinobacteria* (19%, most abundant genus *Corynebacterium*), and while in the POPF group the level of *Firmicutes* was not different (52%), the percentage of *Actinobacteria* was increased (39%, mainly consisting of *Arthrobacter*) and the percentage of *Proteobacteria* (3%) decreased. Looking at *Firmicutes* at the genus level, we found genera such as *Staphylococcus* (52%), *Entercoccus* (18.5%) and *Streptococcus* (12%) in the no POPF group, while in the POPF group the Firmicutes mainly consisted of *Enterococci* (93%). Beta diversity comparisons revealed that patients cluster separately within PCoA (PERMANOVA, Jaccard index, *p*-value < 0.012, F-value 1.8926, R = 0.030). Microbiology reports of the POPF patients were additionally screened, and one of the three patients harbored a multidrug-resistant *Klebsiella oxytoca* and a *Vancomycin-resistant Entercoccus faecium* (VRE) in the drainage fluid (microbial culture two weeks postoperative and the patient received antibiotics). Interestingly, all POPF drainage fluids showed co-colonizations with fungi.

#### 3.5.3. Pancreatic Tumor Tissue, Soft vs. Hard

As the development of a POPF is closely correlated with the texture of the pancreatic tissue, we analyzed the microbial communities of the pancreatic tissue depending on the intraoperative texture of the pancreas, comparing soft (*n* = 4) vs. hard (*n* = 6) pancreases (Figure 7a,b: bar plots representing the relative abundance of bacterial phyla from all sample sites). Therefore, the surgeon documented intraoperatively the pancreas texture. Overall, the soft tissue group harbored more *Firmicutes* (69% vs. hard tissue 29%), while *Proteobacteria* were more abundant in the hard tissue group (44% vs. 9% soft tissue). Beta diversity comparisons revealed that patients cluster separately within PCoA (PERMANOVA, Jaccard index, *p*-value < 0.043, F-value 1.5847, R = 0.025) (Figure 7c). Detailed analysis of the tumor tissue showed that soft tissue harbored more *Fusobacteria* (22% vs. hard tissue 3%), *Proteobacteria* and *Firmicutes*. Hard tissue consisted mainly of *Firmicutes*, *Proteobacteria* and *Verrucomicrobia*. Interestingly, when looking at *Fusobacteria* and POPF formation in soft pancreatic tissue, *Fusobacteria* were more abundant in the no POPF group. In accordance with the recent literature [27], *Fusobacteria* were more abundant in patients with a malignant histology, compared to those with benign lesions.

## 4. Discussion

A growing number of studies have linked the microbiome to carcinogenesis, tumor progression, and therapeutic responses through several pathways, including inflammation, immunity and metabolism, in various cancer types [4,28,29]. Colonization and the diversity of microorganisms in the gastrointestinal tract play an important role in establishing a symbiotic system of host–microbial interactions. Historically, the pancreas and the bile were viewed as sterile compartments, however recent studies have demonstrated that this is not the case [13,30,31].

Little is known about whether local microbiota dysbiosis can affect outcomes in pancreatic cancer surgery. Pancreatic surgery is a high-risk procedure, even in high-volume centers, and POPF formation remains an unsolved problem. Research on the pancreas microbiome has mainly focused on fecal microbiota, mostly because of the convenience of obtaining noninvasive biological samples. Recently, Schmitt et al. correlated the structure of the gut microbiome with the development of postoperative complications after pancreatic surgery. They were able to show that a special bacterial community, which is almost stable during the preoperative and postoperative period, was associated with postoperative complications. This community was characterized by an increase in *Akkermansia*, which belongs to the phyla *Verrucomicrobia* and degrades the intestinal gut mucin. The intestinal mucin layer has an important function as a physiological barrier.

However, prospective studies using solid and liquid biopsies of previously untreated patients for proven or suspected pancreatic cancer are scarce. The goal of this study was to characterize the microbiome within the bile fluid, duodenal mucosa, pancreatic tumor lesions, drainage fluid, and stool samples from patients undergoing pancreatic surgery, and to identify associations between the microbiome profiles and clinical conditions. Various clinically relevant factors can influence the microbiome composition, including antibiotic therapy, proton pump inhibitors (PPIs), and biliary obstruction, which is often treated by endoscopic preoperative biliary stent placement. In addition to the comprehensive evaluation of multiple adjacent pancreas-related compartments, a strength of our study is that our patients did not receive an oral bowel preparation regimen prior to surgery, and the use of solid and liquid biopsies, rather than swabs. We performed an initial exploratory study to test swabs for bio sampling, but there was a large number of swabs that did not generate any data.

This study demonstrates that there is a distinct microbiome in the different compartments adjacent to the pancreas. Three striking patterns emerged: First, the microbiome is altered in patients undergoing preoperative stent placement; second, patients with POPF harbor a distinct microbiome; and third, to the best of our knowledge, so far no study has shown the relationship between the pancreatic gland texture and the microbiome.

The ESMO guidelines recommend that, for patients presenting with jaundice at diagnosis of pancreatic carcinoma, endoscopic drainage preoperatively should only be carried out if there is active cholangitis, or in those for whom curative resection cannot be scheduled within two weeks of diagnosis [32]. Often, however, the indication for ERCP is made even before operability has been determined in an interdisciplinary tumor conference. Several retrospective studies have reported that preoperative biliary drainage is associated with a higher rate of infectious complications after pancreatoduodenectomy [33]. We were able to show that the composition of the bile microbiome and consequently the microbiome of the pancreas and the duodenum markedly differed between patients with or without preoperative stent placement in the common bile duct, most probably due to retrograde bacterial migration along the stent. Due to the small group size (no stent placement *n* = 3), there was no statistically significant difference, but all patients with stent placement had a higher abundance of *Enterococcus*. Furthermore, patients with POPF were more likely to carry bacteria belonging to the genus *Enterococcus*. This is in line with a recent study of biliary smears for routine microbiological diagnostics in pancreatic surgery [34].

Interestingly, we also observed in our cohort *Fusobacteria*, which are linked to carcinogenesis in CRC and pancreatic cancer, in particular *F. nucleatum* [29]. In our cohort Fusobacteria were enriched in malignant pancreatic lesions, which is why we next want to address the pancreatic oncobiome (transformed, dysbiotic microbiome).

Enterococci are intrinsically resistant to cephalosporins. During hospitalization, patients receive standard antibiotics in the preoperative period, and clinicians administer antibiotics to patients with signs of infection or sepsis during the postoperative period. Risk factors associated with pancreatic fistula after pancreatic surgery have been well established. A number of studies have identified and validated that small pancreatic duct size and soft gland texture are the leading risk factors for pancreatic leak following pancreatic surgery [32,35,36,37,38]. That the microbiota contribute to the pathogenesis of pancreatic fistula represents a novel way of thinking about the unsolved POPF problem. We have shown that preoperative biliary stenting influences the local microbiome, and consequently the microbiome of the pancreas, and that dysbiosis affects patient outcome for pancreatic surgery. We therefore believe that a better understanding of the human microbiome will provide exciting opportunities for personalized medicine. Based on our data, a risk adapted selection of the perioperative antibiotic prophylaxis should be seriously considered, and an intraoperative smear should always be taken.

## 5. Conclusions

The microbiome is altered in patients undergoing preoperative stent placement. Most patients undergoing pancreatic surgery for suspected cancer are preoperatively treated with a biliary stent, and this cohort of patients have relatively more *Enterococci* in their bile, tumors, and duodenum. Thus, antibiotic prophylaxis in these patients should have a broad spectrum of coverage for *Enterococci*. On the other hand, a routinely harvested bacterial culture from all patients during diagnostic ERCP for suspected pancreatic cancer, or during the operation, could generate an individual antibiogram, which would be a major support in perioperative treatment in the era of personalized medicine.

Concerning POPF in particular, the role of the microbiome composition may represent one missing piece to explain the unsolved problem of pancreatic fistula development.

## Figures and Tables

**Figure 1 jcm-09-02785-f001:**
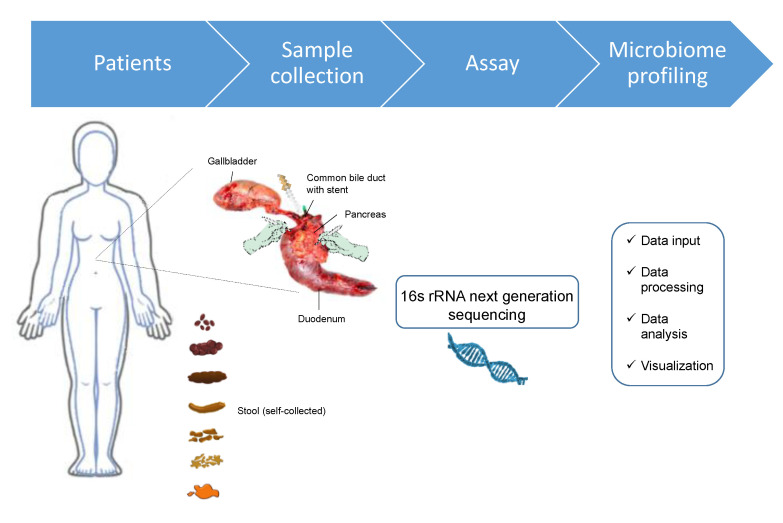
Overview of the study workflow.

**Figure 2 jcm-09-02785-f002:**
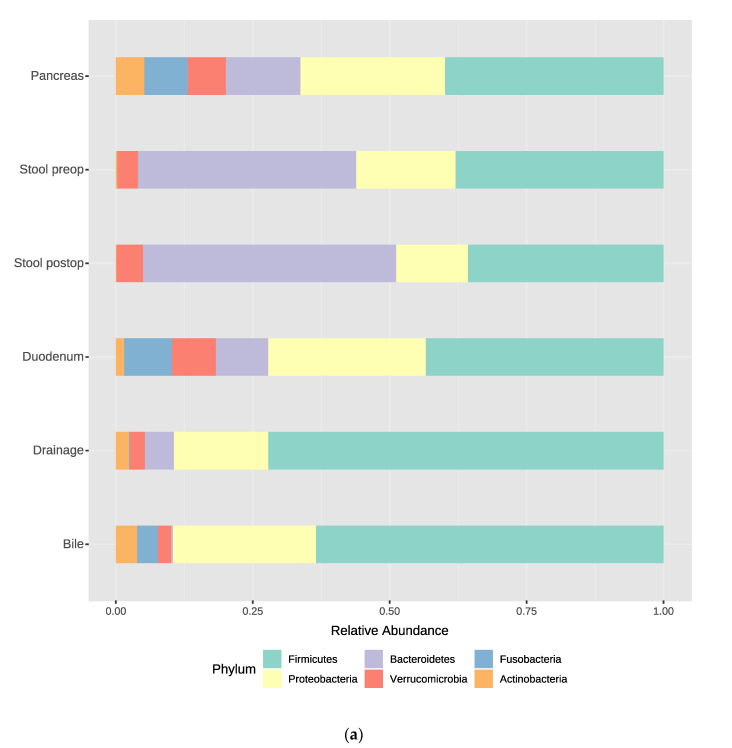
Relative abundance of bacterial taxa in the sampled compartments. (**a**) Taxonomic assignments are shown at the level of the phylum; (**b**) Taxonomic assignments are shown at the level of the genus.

**Figure 3 jcm-09-02785-f003:**
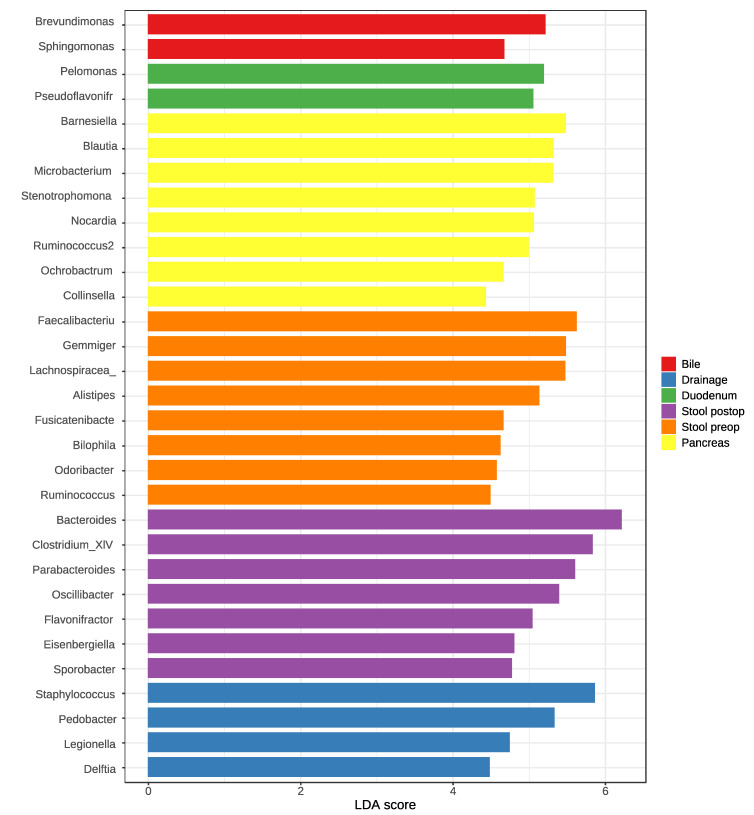
Analysis of key genera of the microbiota in the different body site compartments using linear discriminant analysis (LDA) coupled with effect size measurements (LefSe) analysis.

**Figure 4 jcm-09-02785-f004:**
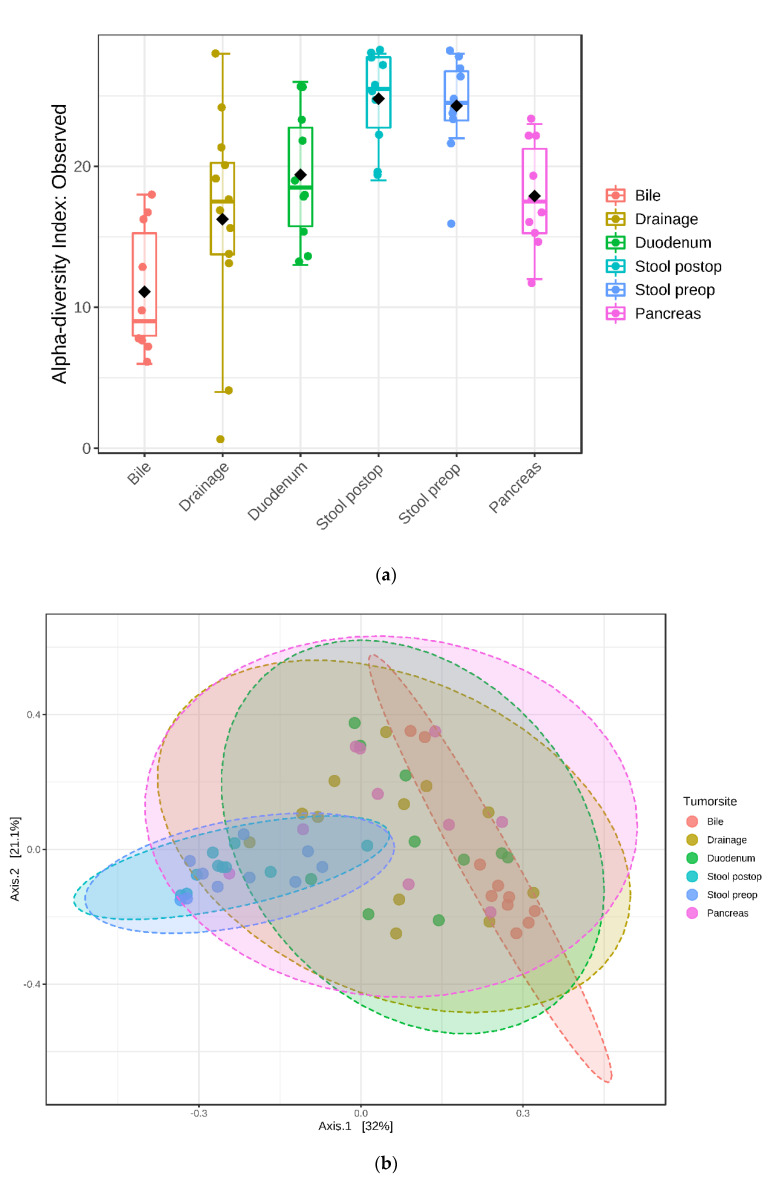
Diversity comparisons of microbial communities. (**a**) alpha diversity index: observed (*p*-value: 4.9602 × 10^−7^; [ANOVA] F-value: 10.29); (**b**) beta diversity PCoA Jensen–Shannon [PERMANOVA] F-value: 4.5694, *p*-value < 0.001.

**Figure 5 jcm-09-02785-f005:**
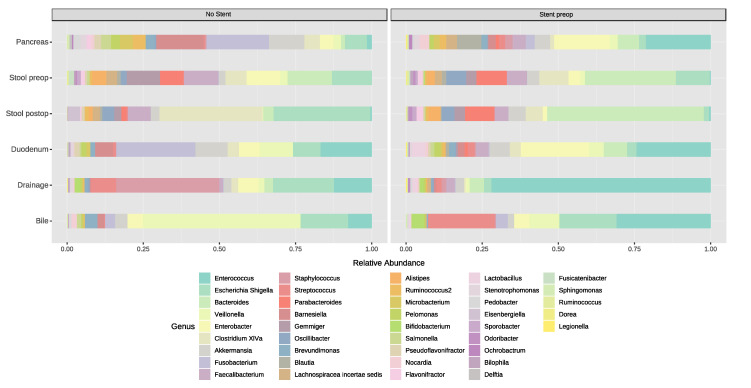
Bar plots representing the microbiome comparison on the genus level of patients without stent placement (no stent) vs. patients with stent placement pre-operative (stent preop).

**Figure 6 jcm-09-02785-f006:**
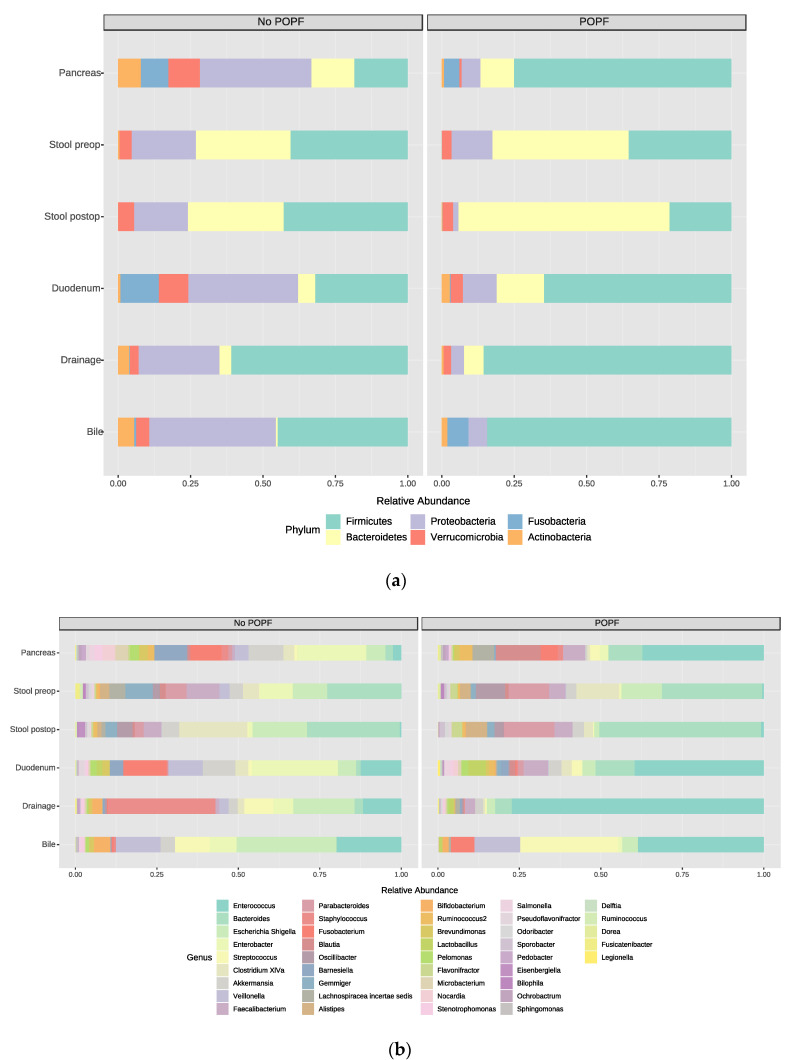
Bar plots representing the microbiome comparison of patients with no POPF (No) vs. patients with clinically relevant POPF (Yes). (**a**) phylum level; (**b**) genus level; (**c**) beta diversity comparison between the two groups shown by PCoA plot (PERMANOVA, Jaccard index, *p*-value < 0.012, F-value 1.8926, R = 0.030) in no POPF (No, red) and POPF (Yes, blue).

**Figure 7 jcm-09-02785-f007:**
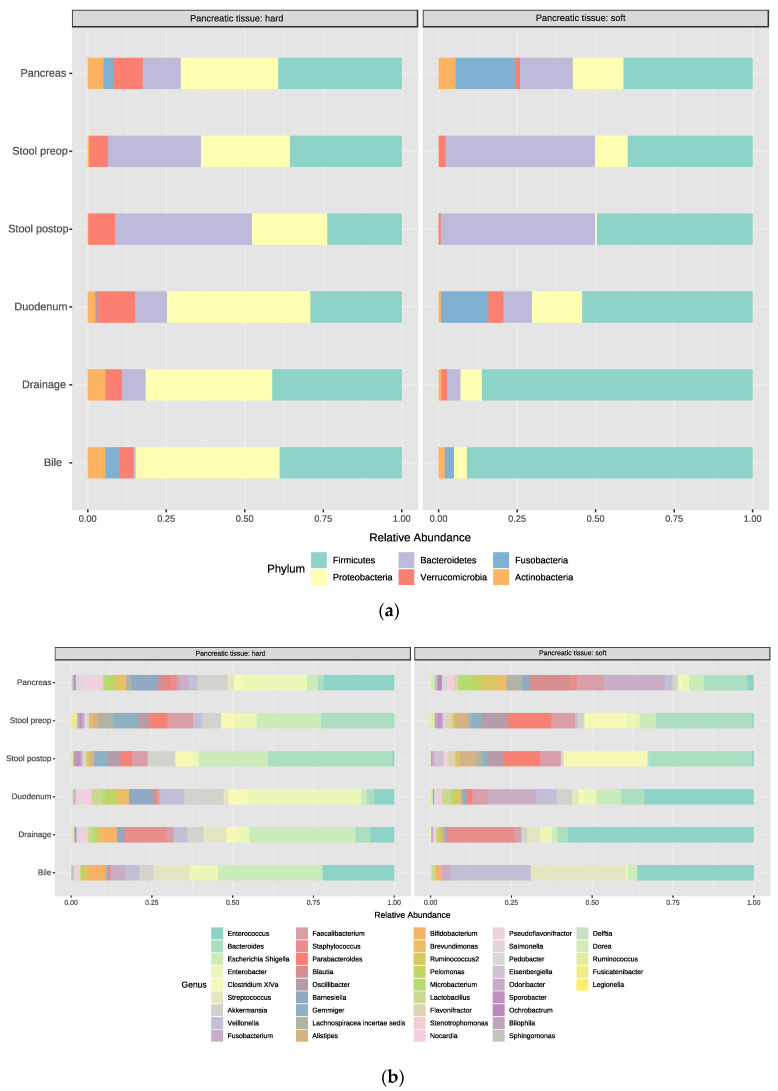
Bar plots representing the relative abundance of the bacterial composition detected in the two tissue types (hard and soft) from all tested sample sites. (**a**) phylum level; (**b**) genus level; (**c**) beta diversity comparison between the two groups shown by PCoA plot (PERMANOVA, Jaccard index, *p*-value < 0.043, F-value 1.5847, R = 0.025) in hard gland texture (Hard, red) and soft gland texture (Soft, blue).

**Table 1 jcm-09-02785-t001:** Clinical characteristics of study participants.

Demographics	Overall (*n* = 10)	POPF	
		Yes (*n* = 3)	No (*n* = 7)
**Age**	44–89	66–79	44–89
**Sex**			
Male	8	3	5
Female	2	0	2
**Body mass index (BMI)**			
18.8–24.9 normal	1	0	1
25.0–29.9 pre obesity	6	3	3
30.0–34.9 obesity class 1	2	0	2
35.0–39.9 obesity class II	1	0	1
**Nicotine**			
Yes	1	0	1
No	9	3	6
**Proton pump inhibitors**			
Yes	4	3	1
No	6	0	6
**Histology**			
Benign	3	1	2
Malignant	7	2	5
**Physical ASA status** (American Society of Anästhesiologists)			
1	0	0	0
2	4	0	4
3	6	3	3
**Stent**			
Yes	7	3	4
No	3	0	3
**Gland texture**			
Soft	4	2	2
Hard	7	1	6
**Diameter of the pancreatic duct**			
1 mm		1 (soft)	1 (hard)
2 mm		0	2
3 mm		0	1
4 mm		1 (hard)	1 (hard)
5 mm		1 (soft)	1 (hard)
6 mm		0	0
7 mm		0	1

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
