# Peer review of "Microbiome Patterns in Matched Bile, Duodenal, Pancreatic Tumor Tissue, Drainage, and Stool Samples: Association with Preoperative Stenting and Postoperative Pancreatic Fistula Development"

_jcm, 2020, doi:10.3390/jcm9092785_

Round 1
Reviewer 1 Report
There are numerous grammatical mistakes in the manuscript that must be corrected.
Reviewer 2 Report
In this manuscript, the authors described microbiome patterns in matched bile, duodenal, pancreatic tumor tissue, drainage, and stool samples association with preoperative stenting and postoperative pancreatic fistula development. It is exciting. However, there are some questions for this manuscript.
- In this manuscript, the authors described patients on biliary stent placement have relatively high amounts of enterococci in the bile, tumor, and duodenum.
- Compared with the condition of obstructive jaundice, the state of placement of a biliary stent is considered to be closer to normal. Nevertheless, what do you think is the cause of the increase in enterococci?
- Also, was the enterococcus the leading cause of postoperative infection in patients with a biliary stent?
- Did the authors distinguish between ENBD and ERBD for indwelling biliary stents?
- In this manuscript, the authors described the presence or absence of POPF due to differences in the gut microbiome. Do the authors think that infection with the increasing bacterias affected POPF formation?
- In this manuscript, the authors were evaluating the microbiome by pancreatic hardness. Are there any indicators of pancreatic hardness?
Reviewer 3 Report
The study aims to characterize and to compare the microbiome of different body sites (bile duct, duodenal mucosa, pancreatic tumor lesion, postoperative drainage fluid, and fecal samples) in patients undergoing pancreatic surgery for suspected pancreatic cancer and correlate those to clinical parameters. The paper is well written. Few comments:
- Why did the authors do not consider the postoperative infectious complications in this series?
- How can the authors explain the finding of MDR bacteria in POPF group? Did they receive antibiotics?
- Which antibiotic prophylaxis used?
- Did the authors have any information about the preoperative stent? Any cholangitis, antibiotic treatment or multiple procedures that can be a confounder
- Did the patients receive any neoadjuvant treatment?
- How do the authors think that these results could change the clinical practice? I have some concerns about it especially if we consider the time of the microbiome analysis. Furthermore, the conclusions on the antibiotic prophylaxis are very optimistic. I don't think should be applicable everywhere.
Round 2
Reviewer 2 Report
Thank you for your revision.
The reviewer has no comment for the revised manuscript.